# Blind Deconvolutional Phase Retrieval via Convex Programming

**Ali Ahmed**
Department of Electrical Engineering
Information Technology University
Lahore, Pakistan.
ali.ahmed@itu.edu.pk

**Alireza Aghasi**
Department of Business Analytics
Georgia State University
Atlanta, GA.
aaghasi@gsu.edu

**Paul Hand**
College of Computer and Information Science
Northeastern University
Boston, MA.
p.hand@northeastern.edu

## Abstract

We consider the task of recovering two real or complex $m$-vectors from phaseless Fourier measurements of their circular convolution. Our method is a novel convex relaxation that is based on a lifted matrix recovery formulation that allows a nontrivial convex relaxation of the bilinear measurements from convolution. We prove that if the two signals belong to known random subspaces of dimensions $k$ and $n$, then they can be recovered up to the inherent scaling ambiguity with $m >> (k + n) \log^2 m$ phaseless measurements. Our method provides the first theoretical recovery guarantee for this problem by a computationally efficient algorithm and does not require a solution estimate to be computed for initialization. Our proof is based Rademacher complexity estimates. Additionally, we provide an ADMM implementation of the method and provide numerical experiments that verify the theory.

## 1    Introduction

This paper considers recovery of two unknown signals (real- or complex-valued) from the magnitude only measurements of their convolution. Let $\boldsymbol{w}$, and $\boldsymbol{x}$ be vectors residing in $\mathcal{H}^m$, where $\mathcal{H}$ denotes either $\mathbb{R}$, or $\mathbb{C}$. Moreover, denote by $\boldsymbol{F}$ the DFT matrix with entries $F[\omega, t] = \frac{1}{\sqrt{m}} \mathrm{e}^{-j2\pi\omega t/m}$, $1 \le \omega, t \le m$. We observe the phaseless Fourier coefficients of the circular convolution $\boldsymbol{w} \circledast \boldsymbol{x}$ of $\boldsymbol{w}$, and $\boldsymbol{x}$

$$\boldsymbol{y} = |\boldsymbol{F}(\boldsymbol{w} \circledast \boldsymbol{x})|, \tag{1}$$

where $|\boldsymbol{z}|$ returns the element wise absolute value of the vector $\boldsymbol{z}$. We are interested in recovering $\boldsymbol{w}$, and $\boldsymbol{x}$ from the phaseless measurements $\boldsymbol{y}$ of their circular convolution. In other words, the problem concerns blind deconvolution of two signals from phaseless measurements. The problem can also be viewed as identifying the structural properties on $\boldsymbol{w}$ such that its convolution with the signal/image of interest $\boldsymbol{x}$ makes the phase retrieval of a signal $\boldsymbol{x}$ well-posed. Since $\boldsymbol{w}$, and $\boldsymbol{x}$ are both unknown, and in addition, the measurements are phaseless, the inverse problem becomes severly ill-posed as many pairs of $\boldsymbol{w}$, and $\boldsymbol{x}$ correspond to the same $\boldsymbol{y}$. We show that this non-linear problem can be efficiently solved, under Gaussian measurements, using a semidefinite program and also theoretically prove this assertion. We also propose a heuristic approach to solve the proposed

semidefinite program computationally efficiently. Numerical experiments show that, using this algorithm, one can successfully recover a blurred image from the magnitude only measurements of its Fourier spectrum.

Phase retrieval has been of continued interest in the fields of signal processing, imaging, physics, computational science, etc. Perhaps, the single most important context in which phase retrieval arises is the X-ray crystallography Harrison [1993], Millane [1990], where the far-field pattern of X-rays scattered from a crystal form a Fourier transform of its image, and it is only possible to measure the intensities of the electromagnetic radiation. However, with the advancement of imaging technologies, the phase retrieval problem continues to arise in several other imaging modalities such as diffraction imaging Bunk et al. [2007], microscopy Miao et al. [2008], and astronomical imagingFienup and Dainty [1987]. In the imaging context, the result in this paper would mean that if rays are convolved with a *generic* pattern (either man made or naturally arising due to propagation of light through some unknown media) $w$ prior to being scattered/reflected from the object, the image of the object can be recovered from the Fourier intensity measurements later on. As is well known from Fourier optics Goodman [2008], the convolution of a visible light with a generic pattern can be implemented using a lens-grating-lens setup.

Despite recent advances in theoretical understanding of phase retrieval Candes et al. [2013, 2015a], the application to actual problems such as crystallography remains challenging owing partly to the simplistic mathematical models that may not fully capture the actual physical problem at hand. Our comparatively more complex model in (1) more elaborately encompasses structure in actual physical problems, for example, crystallography, where due to the natural periodic arrangement of a crystal structural unit, the observed electron density function of the crystal exactly takes the form (1); for details, see, Section 2 of Elser et al. [2017].

Blind deconvolution is a fundamental problem in signal processing, communications, and in general system theory. Visible light communication has been proposed as a standard in 5G communications for local area networks Azhar et al. [2013], Retamal et al. [2015], Azhar et al. [2010]. Propagation of information carrying light through an unknown communication medium is modeled as a convolution. The channel is unknown and at the receiver it is generally difficult to measure the phase information in the propagated light. The result in this paper says that the transmitted signal can be blindly deconvolved from the unknown channel from the Fourier intensity measurements of the light only. The reader is referred to the first section of the supplementary note for a detailed description of the visible light communication and its connection to our formulation.

## 1.1 Observations in Matrix Form

The phase retrieval, and blind deconvolution problem has been extensively studied in signal processing community in recent years Candes et al. [2015b], Ahmed et al. [2014] by lifting the unknown vectors to a higher dimensional matrix space formed by their outer products. The resulting rank-1 matrix is recovered using nuclear norm as a convex relaxation of the non-convex rank constraint. Recently, other forms of convex relaxations have been proposed Bahmani and Romberg [2017a], Goldstein and Studer [2018], Aghasi et al. [2017a,b] that solve both the problems in the native (unlifted) space leading to computationally efficiently solvable convex programs. This paper handles the non-linear convolutional phase retrieval problem by lifting it into a bilinear problem. The resulting problem, though still non-convex, gives way to an effective convex relaxation that provably recovers $w$, and $x$ exactly.

It is clear from (1) that uniquely recovering $w$, and $x$ is not possible without extra knowledge or information about the problem. We will address the problem under a broad and generally applicable structural assumptions that both the vectors $w$, and $x$ are members of known subspaces of $\mathcal{H}^m$. This means that $w$, and $x$ can be parameterized in terms of unknown lower dimensional vectors $h \in \mathcal{H}^k$, and $m \in \mathcal{H}^n$, respectively as follows

$$w = Bh, \ x = Cm, \tag{2}$$

where $B \in \mathcal{H}^{m \times k}$, and $C \in \mathcal{H}^{m \times n}$ are known matrices whose columns span the subspaces in which $w$, and $x$ reside, respectively. Recovering $h$, and $m$ would imply the recovery of $w$, and $x$, therefore, we take $h$, and $m$ as the unknowns in the inverse problem henceforth. Since the circular convolution operator diagonalizes in the Fourier domain, the measurements in (1) take the following form after

incorporating the subspace constraints in (2)

$$y = \tfrac{1}{\sqrt{m}}|\hat{B}h \odot \hat{C}m|,$$

where $\hat{B} = \sqrt{m}FB$, $\hat{C} = \sqrt{m}FC$, and $\odot$ represent the Hadamard product. Denoting by $b_\ell^*$ and $c_\ell^*$ the rows of $\hat{B}$, and $\hat{C}$, respectively, the entries of the measurements $y$ can be expressed as

$$y_\ell^2 = \tfrac{1}{m}|\langle b_\ell, h\rangle\langle c_\ell, m\rangle|^2, \; \ell = 1, 2, 3, \ldots, m.$$

Evidently the problem is non-linear in both unknowns. However, it reduces to a bilinear problem in the lifted variables $hh^*$, and $mm^*$ taking the form

$$y_\ell^2 = \tfrac{1}{m}\langle b_\ell b_\ell^*, hh^*\rangle\langle c_\ell c_\ell^*, mm^*\rangle = \tfrac{1}{m}\langle b_\ell b_\ell^*, H\rangle\langle c_\ell c_\ell^*, M\rangle, \tag{3}$$

where $H$, and $M$ are the rank-1 matrices $hh^*$, and $mm^*$, respectively. Treating the lifted variables $H$, and $M$ as unknowns makes the measurements bilinear in the unknowns; a structure that will help us formulate an effective convex relaxation.

## 1.2 Novel Convex Relaxation

The task of recovering $H$, and $M$ from $y$ in (3) can be naturally posed as an optimization program

$$\text{find } H, M \tag{4}$$
$$\text{subject to } \tfrac{1}{m}\langle b_\ell b_\ell^*, H\rangle\langle c_\ell c_\ell^*, M\rangle = y_\ell^2, \; \ell = 1, 2, 3, \ldots, m.$$
$$\text{rank}(H) = 1, \; \text{rank}(M) = 1.$$

However, both the measurement and the rank constraints are non-convex. Further, the immediate convex relaxation of each measurement constraint is trivial, as the convex hull of the set of $(H, M)$ satisfying $y_\ell^2 = \tfrac{1}{m}\langle b_\ell b_\ell^*, H\rangle\langle c_\ell c_\ell^*, M\rangle$ is the set of all possible $(H, M)$.

To derive our convex relaxation, recall that the true $H = hh^*$, and $M = mm^*$ are also positive semidefinite (PSD). This means that incorporating the PSD constraint in the optimization program translates into the fact that the variables $u_\ell = \langle b_\ell b_\ell^*, H\rangle$ and $v_\ell = \langle c_\ell c_\ell^*, M\rangle$ are necessarily non-negative. That is,

$$H \succcurlyeq 0, \text{ and } M \succcurlyeq 0 \implies u_\ell \geq 0, \text{ and } v_\ell \geq 0,$$

where the implication simply follows by the definition of PSD matrices. This observation restricts the hyperbolic constraint set in Figure 1 to the first quadrant only. For a fixed $\ell$, we propose replacing the non-convex hyperbolic set $\{(u_\ell, v_\ell) \in \mathbb{R}^2 \mid \tfrac{1}{m}u_\ell v_\ell = y_\ell^2, u_\ell \geq 0, \; v_\ell \geq 0\}$ with its convex hull $\{(u_\ell, v_\ell) \in \mathbb{R}^2 \mid \tfrac{1}{m}u_\ell v_\ell \geq y_\ell^2, u_\ell \geq 0, \; v_\ell \geq 0\}$. In short, our convex relaxation is possible because the PSD constraint from lifting happens to select a specific branch of the hyperbola given by any particular bilinear measurement, and this single branch has a nontrivial convex hull.

The rest of the convex relaxation is standard, as the rank constraint in (4) is then relaxed with a nuclear-norm minimization, which reduces to trace minimization in the PSD case. Hence, we study the convex program

$$\text{minimize } \text{Tr}(H) + \text{Tr}(M) \tag{5}$$
$$\text{subject to } \tfrac{1}{m}\langle b_\ell b_\ell^*, H\rangle\langle c_\ell c_\ell^*, M\rangle \geq y_\ell^2, \; \ell = 1, 2, \ldots, m$$
$$H \succcurlyeq 0, \; M \succcurlyeq 0.$$

The following lemma formally proves the convexity of the optimization program above.

**Lemma 1.** *If $y \in \mathbb{R}^m$ such that $y_\ell > 0$ then the optimization program in (5) is a convex program.*

*Proof.* The objective of (5) is simply linear, we focus on the constraints. For a fixed $\ell$, let $S_\ell := \{(H, M) \in \mathcal{H}^{k\times k} \times \mathcal{H}^{m\times m} \mid \tfrac{1}{m}\langle b_\ell b_\ell^*, H\rangle\langle c_\ell c_\ell^*, M\rangle \geq y_\ell^2, H \succcurlyeq 0, M \succcurlyeq 0\}$, $S_{\ell,1} := \{(u_\ell, v_\ell) \in \mathbb{R}^2 \mid \tfrac{1}{m}u_\ell v_\ell \geq y_\ell^2, u_\ell \geq 0, v_\ell \geq 0\}$, and $S_{\ell,2} := \{(H, M) \in \mathcal{H}^{k\times k} \times \mathcal{H}^{m\times m} \mid (\langle b_\ell b_\ell^*, H\rangle, \langle c_\ell c_\ell^*, M\rangle) \in S_{\ell,1}\}$. To show that $S_\ell$ is convex, it suffices to show that $S_{\ell,1}$, and $S_{\ell,2}$ are convex.

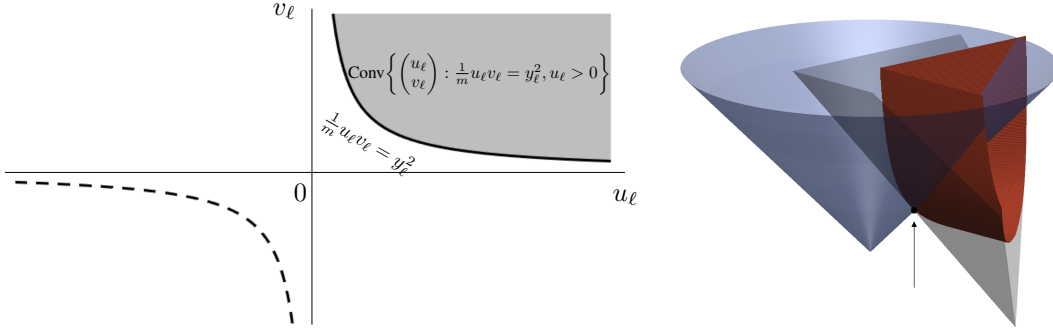

Figure 1: Left: Restriction of the hyperbolic constraint to the first quadrant; Right: Abstract Illustration of the Geometry of the Convex Relaxation. PSD cone (blue) and the surface of the hyperbolic set (red) formed by two intersecting hyperbolas ($m = 2$). Evidently, there are multiple points on the surface and also in the convex hull of the hyperbolic set that lie on the PSD cone. The minimizer of the optimization program (5) picks the one with minimum trace that happens to lie at the intersection of hyperbolic ridge and the PSD cone (pointed out by an arrow). The gray envelope of two ($m = 2$) hyperplanes surrounding the hyperbolic set correspond to the linearization of the hyperbolic set at the minimizer; this forms the basis of a connected linearly constrained program later in (9).

Fix $(u_1, v_1), (u_2, v_2) \in S_{\ell,1}$, and let $\alpha \in [0, 1]$. Note that $u_1 > 0$, and $u_2 > 0$ as $y_\ell > 0$. Consider

$$\frac{1}{m}(\alpha u_1 + (1 - \alpha)u_2)(\alpha v_1 + (1 - \alpha)v_2)$$

$$= \frac{1}{m}\left((\alpha^2 u_1 v_1 + (1 - \alpha)^2 u_2 v_2) + \alpha(1 - \alpha)(u_1 v_2 + u_2 v_1)\right)$$

$$\geq (\alpha^2 y_\ell^2 + (1 - \alpha)^2 y_\ell^2) + \alpha(1 - \alpha)(\frac{y_\ell^2 u_1}{u_2} + \frac{y_\ell^2 u_2}{u_1})$$

$$= y_\ell^2 \left(1 + \frac{2\alpha^2 u_1 u_2 - 2\alpha u_1 u_2 + \alpha(1 - \alpha)(u_1^2 + u_2^2)}{u_1 u_2}\right)$$

$$= y_\ell^2 \left(1 + \frac{(\alpha - \alpha^2)(u_1 - u_2)^2}{u_1 u_2}\right) \geq y_\ell^2,$$

where the last inequality follows form the fact that $\alpha \in [0, 1]$, and $u_1 u_2 > 0$. This shows that $S_{\ell,1}$ is convex.

The set $S_{\ell,2}$ is convex as the inverse image of a convex set of a linear map is convex. This implies that $S_\ell$ is convex. Finally, the intersection of any number of convex sets is convex means that the constraint of (5) is convex. This proves that (5) is a convex program. □

## 1.3 Main Result

As we are presenting the first analytical results on this problem, we choose the subspace matrices $\boldsymbol{B}$, and $\boldsymbol{C}$ to be standard Gaussian:

$$B[\ell, i] \sim \text{Normal}(0, \tfrac{1}{m}), (\ell, i) \in [m] \times [k], \text{and } C[\ell, i] \sim \text{Normal}(0, \tfrac{1}{m}), (\ell, i) \in [m] \times [n]. \quad (6)$$

Note that this choice results in $\boldsymbol{b}_\ell, \boldsymbol{c}_\ell \sim \text{Normal}(\boldsymbol{0}, \boldsymbol{I})$. We show that with this choice the optimization program in (5) recovers a global scaling of $(\alpha \boldsymbol{H}^\natural, \alpha^{-1} \boldsymbol{M}^\natural)$ of the true solution $(\boldsymbol{H}^\natural, \boldsymbol{M}^\natural)$. We will interchangeably use the notation $(\boldsymbol{H}, \boldsymbol{M}) \in (\mathcal{H}^{k \times k}, \mathcal{H}^{n \times n})$ to denote the pair of matrices $\boldsymbol{H}$ and $\boldsymbol{M}$, or the block diagonal matrix

$$(\boldsymbol{H}, \boldsymbol{M}) = \begin{bmatrix} \boldsymbol{H} & \boldsymbol{0} \\ \boldsymbol{0} & \boldsymbol{M} \end{bmatrix}. \quad (7)$$

The exact value of the unknown scalar multiple $\alpha$ can be characterized for the solution of (5). Observe that the solution $(\widehat{\boldsymbol{H}}, \widehat{\boldsymbol{M}})$ of the convex optimization program in (5) obeys $\mathrm{Tr}(\widehat{\boldsymbol{H}}) = \mathrm{Tr}(\widehat{\boldsymbol{M}})$. We aim to show that the solution of the optimization program recovers the scaling $(\tilde{\boldsymbol{H}}, \tilde{\boldsymbol{M}})$ of the true solution $(\boldsymbol{H}^\natural, \boldsymbol{M}^\natural)$:

$$\tilde{\boldsymbol{H}} = \sqrt{\frac{\mathrm{Tr}(\boldsymbol{M}^\natural)}{\mathrm{Tr}(\boldsymbol{H}^\natural)}}\boldsymbol{H}^\natural, \; \tilde{\boldsymbol{M}} = \sqrt{\frac{\mathrm{Tr}(\boldsymbol{H}^\natural)}{\mathrm{Tr}(\boldsymbol{M}^\natural)}}\boldsymbol{M}^\natural. \tag{8}$$

Note that $\mathrm{Tr}(\tilde{\boldsymbol{H}}) = \mathrm{Tr}(\tilde{\boldsymbol{M}})$. The main result can now be stated as follows.

**Theorem 1** (Exact Recovery). *Given the magnitude only spectrum measurements* (1) *of the convolution of two unknown vectors $\boldsymbol{w}^\natural$, and $\boldsymbol{x}^\natural$ in $\mathcal{H}^m$. Suppose that $\boldsymbol{w}^\natural$, and $\boldsymbol{x}^\natural$ are generated as in* (2), *where $\boldsymbol{B}$, and $\boldsymbol{C}$ are known standard Gaussian matrices as in* (6). *Then the convex optimization program in* (5) *uniquely recovers $(\alpha\boldsymbol{H}^\natural, \alpha^{-1}\boldsymbol{M}^\natural)$ for $\alpha = \sqrt{\frac{\mathrm{Tr}\,\boldsymbol{M}^\natural}{\mathrm{Tr}\,\boldsymbol{H}^\natural}}$ with probability at least $1 - \exp(-\frac{1}{2}mt^2)$ whenever $m \geq c\left(\sqrt{(k+n)}\log m + t\right)^2$, where $c$ is an absolute constant.*

## 1.4 Main Contributions

In this paper, we study the combination of two important and notoriously challenging signal recovery problems: phase retrieval and blind deconvolution. We introduce a novel convex formulation that is possible because the algebraic structure from lifting resolves the bilinear ambiguity just enough to permit a nontrivial convex relaxation of the measurements. The strengths of our approach are that it allows a novel convex program that is the first to provably permit recovery guarantees with optimal sample complexity for the joint task of phase retrieval and blind deconvolution when the signals belong to known random subspaces. Additionally, unlike many recent convex relaxations and nonconvex approaches, our approach does not require an initialization or estimate of the true solution in order to be stated or solved. Admittedly, our method, directly interpreted, is computationally prohibitive for large problem sizes because lifting squares the dimensionality of the problem. Nonetheless, techniques, such as Burer-Monteiro approaches that only maintain low-rank representations Burer and Monteiro [2003], have been developed for similar problems. This current work provides the theoretical justification for the exploration of such problems in this difficult combination of phase retrieval and blind deconvolution, and we leave such work for future research.

We do not want to give the reader the impression that the present paper solves the problem of blind deconvolutional phase retrieval in practice. The numerical experiments we perform do indeed show excellent agreement with the theorem in the case of random subspaces. Such subspaces are unlikely to appear in practice, and typically appropriate subspaces would be deterministic, including partial Discrete Cosine Transforms or partial Discrete Wavelet Transforms. Numerical experiments, not shown, indicate that our convex relaxation is less effective for the cases of these deterministic subspaces. We suspect this is due to the fact that the subspaces for both measurements should be mutually incoherent, in addition to both being incoherent with respect to the Fourier basis given by the measurements. As with the initial recovery theory for the problems of compressed sensing and phase retrieval, we have studied the random case in order to show information theoretically optimal sample complexity is possible by efficient algorithms. Based on this work, it is clear that blind deconvolutional phase retrieval is still a very challenging problem in the presence of deterministic matrices, and one for which development of convex or nonconvex methods may provide substantial progress in applications.

## 2 Proof of Theorem 1

To prove Theorem 1, we will show that $(\tilde{\boldsymbol{H}}, \tilde{\boldsymbol{M}})$ is the unique minimizer of an optimization program with a larger feasible set defined by linear constraints.

**Lemma 2.** *If $(\tilde{\boldsymbol{H}}, \tilde{\boldsymbol{M}})$ is the unique solution to*

$$\text{minimize } \|\boldsymbol{H}\|_* + \|\boldsymbol{M}\|_* \tag{9}$$
$$\text{subject to } \frac{1}{m}(\langle \boldsymbol{b}_\ell \boldsymbol{b}_\ell^*, \boldsymbol{H}\rangle\langle \boldsymbol{c}_\ell \boldsymbol{c}_\ell^*, \tilde{\boldsymbol{M}}\rangle + \langle \boldsymbol{b}_\ell \boldsymbol{b}_\ell^*, \tilde{\boldsymbol{H}}\rangle\langle \boldsymbol{c}_\ell \boldsymbol{c}_\ell^*, \boldsymbol{M}\rangle) \geq 2y_\ell^2,$$
$$\ell = 1, 2, 3, \dots, m.$$

*then $(\tilde{\boldsymbol{H}}, \tilde{\boldsymbol{M}})$ is the unique solution to (5).*

*Proof.* Start by observing that the trace in (5) can be replaced with nuclear norm as on the set of PSD matrices both are equivalent. This gives

$$\text{minimize } \|\boldsymbol{H}\|_* + \|\boldsymbol{M}\|_* \qquad (10)$$
$$\text{subject to } \tfrac{1}{m}\langle \boldsymbol{b}_\ell \boldsymbol{b}_\ell^*, \boldsymbol{H}\rangle\langle \boldsymbol{c}_\ell \boldsymbol{c}_\ell^*, \boldsymbol{M}\rangle \geq y_\ell^2, \ \ell = 1, 2, \ldots, m$$
$$\boldsymbol{H} \succcurlyeq 0, \ \boldsymbol{M} \succcurlyeq 0.$$

It suffices now to show that the feasible set of (9) contains the feasible set of (10). Recall the notations

$$u_\ell = \langle \boldsymbol{b}_\ell \boldsymbol{b}_\ell^*, \boldsymbol{H}\rangle, \ v_\ell = \langle \boldsymbol{c}_\ell \boldsymbol{c}_\ell^*, \boldsymbol{M}\rangle, \ \tilde{u}_\ell = \langle \boldsymbol{b}_\ell \boldsymbol{b}_\ell^*, \tilde{\boldsymbol{H}}\rangle, \text{ and } \tilde{v}_\ell = \langle \boldsymbol{c}_\ell \boldsymbol{c}_\ell^*, \tilde{\boldsymbol{M}}\rangle.$$

Using the fact that a convex set with smooth boundary is contained in a half space defined by the tangent hyperplane at any point on the boundary of the set. Consider the point $(\tilde{u}_\ell, \tilde{v}_\ell) \in \mathbb{R}^2$, and observe that

$$\left\{(u_\ell, v_\ell) \in \mathbb{R}^2 \mid \tfrac{1}{m}u_\ell v_\ell \geq y_\ell^2, u_\ell \geq 0, \text{ and } v_\ell \geq 0\right\} \subseteq \left\{(u_\ell, v_\ell) \in \mathbb{R}^2 \mid \tfrac{1}{m}\begin{bmatrix}\tilde{v}_\ell \\ \tilde{u}_\ell\end{bmatrix} \cdot \begin{bmatrix}u_\ell - \tilde{u}_\ell \\ v_\ell - \tilde{v}_\ell\end{bmatrix} \geq 0\right\}.$$

Rewriting $u_\ell$ and $v_\ell$ in the form of original constraints, we have that any feasible point $(\tilde{\boldsymbol{H}}, \tilde{\boldsymbol{M}})$ of (10) satisfies $\tfrac{1}{m}(\langle \boldsymbol{b}_\ell \boldsymbol{b}_\ell^*, \boldsymbol{H}\rangle\langle \boldsymbol{c}_\ell \boldsymbol{c}_\ell^*, \tilde{\boldsymbol{M}}\rangle + \langle \boldsymbol{b}_\ell \boldsymbol{b}_\ell^*, \tilde{\boldsymbol{H}}\rangle\langle \boldsymbol{c}_\ell \boldsymbol{c}_\ell^*, \boldsymbol{M}\rangle) \geq 2y_\ell^2, \ \ell = 1, 2, 3, \ldots, m.$ □

The geometry of the linearly constrained program (9) is also shown in Figure 1 (Right), where the hyperbolic set is replaced by an envelop of hyperplanes defined by the linear constraints of (9). Visually it is clear from Figure 1 that the feasible set of (9) is larger than that of (5).

Define a set $\mathcal{S} := \{(\boldsymbol{H}, \boldsymbol{M}) \mid (\boldsymbol{H}, \boldsymbol{M}) = \alpha(-\tilde{\boldsymbol{H}}, \tilde{\boldsymbol{M}}), \text{ and } \alpha \in [-1, 1]\}$, and $\boldsymbol{A}_\ell = (\tilde{v}_\ell \boldsymbol{b}_\ell \boldsymbol{b}_\ell^*, \tilde{u}_\ell \boldsymbol{c}_\ell \boldsymbol{c}_\ell^*) \in \mathcal{H}^{(k+n)\times(k+n)}$, and define a linear map $\mathcal{A} : \mathcal{H}^{(k+n)\times(k+n)} \to \mathcal{H}^m$ as $\mathcal{A}((\boldsymbol{H}, \boldsymbol{M})) = [\langle \boldsymbol{A}_1, (\boldsymbol{H}, \boldsymbol{M})\rangle, \ldots, \langle \boldsymbol{A}_m, (\boldsymbol{H}, \boldsymbol{M})\rangle]^\mathsf{T}$; one can imagine $\mathcal{A}$ as a matrix with vectorized $\boldsymbol{A}_\ell$ as its rows. The linear constraints in the (9) are $\mathcal{A}((\boldsymbol{H}, \boldsymbol{M})) \geq 2\boldsymbol{y}^2$; the inequality here applies elementwise. Furthermore, define $\mathcal{N} := \text{span}((-\tilde{\boldsymbol{H}}, \tilde{\boldsymbol{M}}))$, and it is easy to see that $\mathcal{S} \subset \mathcal{N} \subseteq \text{Null}(\mathcal{A})$.

We want to show that any feasible perturbation $(\delta\boldsymbol{H}, \delta\boldsymbol{M})$ around the truth $(\tilde{\boldsymbol{H}}, \tilde{\boldsymbol{M}})$ strictly increases the objective. From the discussion above, it is clear that the perturbations $(\delta\boldsymbol{H}, \delta\boldsymbol{M}) \in \mathcal{S}$ do not change the objective and also lead to feasible points of (9). Our general strategy will be to resolve any perturbation $(\delta\boldsymbol{H}, \delta\boldsymbol{M})$ into two components, one in $\mathcal{N}$ and the other in $\mathcal{N}_\perp$, where $\mathcal{N}_\perp$ is the orthogonal complement of the subspace $\mathcal{N}$. The component in $\mathcal{N}$ does not affect the objective. We show that the components in $\mathcal{N}_\perp$ of all the feasible perturbations lead to a strict increase in the objective of (9). This should imply that that the minimizer of (9) can be anywhere in the set[1] $(\tilde{\boldsymbol{H}}, \tilde{\boldsymbol{M}}) \oplus \mathcal{N}$. However, as we are minimizing the (trace) norms, an arbitrary large scaling of the solution is prevented and it is restricted to the subset $(\tilde{\boldsymbol{H}}, \tilde{\boldsymbol{M}}) \oplus \mathcal{S}$. Moreover, among these solutions only $(\tilde{\boldsymbol{H}}, \tilde{\boldsymbol{M}})$ lies in the feasible set of (10). Given this and the fact that $(\tilde{\boldsymbol{H}}, \tilde{\boldsymbol{M}})$ is a minimizer of (9) implies that $(\tilde{\boldsymbol{H}}, \tilde{\boldsymbol{M}})$ is the unique minimizer of (10).

We begin by characterizing the set of descent directions for the objective function of the optimization program (9). Let $T_{\tilde{h}}$, and $T_{\tilde{m}}$ be the set of symmetric matrices of the form

$$T_{\tilde{h}} := \{\boldsymbol{X} = \tilde{\boldsymbol{h}}\boldsymbol{z}^* + \boldsymbol{z}\tilde{\boldsymbol{h}}^*\}, \ T_{\tilde{m}} := \{\boldsymbol{X} = \tilde{\boldsymbol{m}}\boldsymbol{z}^* + \boldsymbol{z}\tilde{\boldsymbol{m}}^*\},$$

and denote the orthogonal complements by $T_{\tilde{h}}^\perp$, and $T_{\tilde{m}}^\perp$, respectively. Note that $\boldsymbol{X} \in T_{\tilde{h}}^\perp$ iff both the row and column spaces of $\boldsymbol{X}$ are perpendicular to $\tilde{\boldsymbol{h}}$. $\mathcal{P}_{T_{\tilde{h}}}$ denotes the orthogonal projection onto the set $T_{\tilde{h}}$, and a matrix $\boldsymbol{X}$ of appropriate dimensions can be projected into $T_{\tilde{h}}$ as

$$\mathcal{P}_{T_{\tilde{h}}}(\boldsymbol{X}) := \frac{\tilde{\boldsymbol{h}}\tilde{\boldsymbol{h}}^*}{\|\tilde{\boldsymbol{h}}\|_2^2}\boldsymbol{X} + \boldsymbol{X}\frac{\tilde{\boldsymbol{h}}\tilde{\boldsymbol{h}}^*}{\|\tilde{\boldsymbol{h}}\|_2^2} - \frac{\tilde{\boldsymbol{h}}\tilde{\boldsymbol{h}}^*}{\|\tilde{\boldsymbol{h}}\|_2^2}\boldsymbol{X}\frac{\tilde{\boldsymbol{h}}\tilde{\boldsymbol{h}}^*}{\|\tilde{\boldsymbol{h}}\|_2^2}$$

Similarly, define the projection operator $\mathcal{P}_{T_{\tilde{m}}}$. The projection onto orthogonal complements are then simply $\mathcal{P}_{T_{\tilde{h}}^\perp} := \mathcal{I} - \mathcal{P}_{T_{\tilde{h}}}$, and $\mathcal{P}_{T_{\tilde{m}}^\perp} := \mathcal{I} - \mathcal{P}_{T_{\tilde{m}}}$, where $\mathcal{I}$ is the identity operator. We use $\boldsymbol{X}_{T_{\tilde{h}}}$ as a

shorthand for $\mathcal{P}_{T_{\tilde{h}}}(\boldsymbol{X})$. Using the notation in (7), the objective of (9) is $\|(\boldsymbol{H}, \boldsymbol{M})\|_*$, and subgradient of the objective at the proposed solution $(\tilde{\boldsymbol{H}}, \tilde{\boldsymbol{M}})$ is

$$\partial\|(\tilde{\boldsymbol{H}}, \tilde{\boldsymbol{M}})\|_* := \big\{ \boldsymbol{G} = (\tilde{\boldsymbol{h}}\tilde{\boldsymbol{h}}^*, \tilde{\boldsymbol{m}}\tilde{\boldsymbol{m}}^*) + (\boldsymbol{W}_{T_{\tilde{h}}^\perp}, \boldsymbol{W}_{T_{\tilde{m}}^\perp}), \ \|(\boldsymbol{W}_{T_{\tilde{h}}^\perp}, \boldsymbol{W}_{T_{\tilde{m}}^\perp})\| \le 1 \big\}.$$

The set $\mathcal{Q}$ of descent directions of the objective of (9) is defined as

$$\big\{ (\delta\boldsymbol{H}, \delta\boldsymbol{M}) \in \mathcal{N}_\perp : \big\langle (\boldsymbol{G}, (\delta\boldsymbol{H}, \delta\boldsymbol{M}) \big\rangle \le 0, \forall \boldsymbol{G} \in \partial\|(\tilde{\boldsymbol{H}}, \tilde{\boldsymbol{M}})\|_* \big\} \subseteq$$

$$\big\{ (\delta\boldsymbol{H}, \delta\boldsymbol{M}) \in \mathcal{N}_\perp : \big\langle (\tilde{\boldsymbol{h}}\tilde{\boldsymbol{h}}^*, \tilde{\boldsymbol{m}}\tilde{\boldsymbol{m}}^*), (\delta\boldsymbol{H}, \delta\boldsymbol{M}) \big\rangle +$$

$$\|(\delta\boldsymbol{H}_{T_{\tilde{h}}^\perp}, \delta\boldsymbol{M}_{T_{\tilde{m}}^\perp})\|_* \le 0, \forall \boldsymbol{G} \in \partial\|(\tilde{\boldsymbol{H}}, \tilde{\boldsymbol{M}})\|_* \big\} \subset$$

$$\big\{ (\delta\boldsymbol{H}, \delta\boldsymbol{M}) \in \mathcal{N}_\perp : \|(\delta\boldsymbol{H}_{T_{\tilde{h}}^\perp}, \delta\boldsymbol{M}_{T_{\tilde{m}}^\perp})\|_* \le \|(\delta\boldsymbol{H}_{T_{\tilde{h}}}, \delta\boldsymbol{M}_{T_{\tilde{m}}})\|_F, \ \forall \boldsymbol{G} \in \partial\|(\tilde{\boldsymbol{H}}, \tilde{\boldsymbol{M}})\|_* \big\}$$

$$=: \mathcal{Q}. \tag{11}$$

We quantify the "width" of the set of descent directions $\mathcal{Q}$ through a Rademacher complexity, and a probability that the gradients of the constraint functions of (9) lie in a certain half space. This enables us to build an argument using the small ball method Koltchinskii and Mendelson [2015], Mendelson [2014] that it is unlikely to have points that meet the constraints in (9) and still be in $\mathcal{Q}$. Before moving forward, we introduce the above mentioned Rademacher complexity and probability term.

Denote the constraint functions as[2] $f_\ell(\boldsymbol{H}, \boldsymbol{M}) = \tilde{u}_\ell \langle \boldsymbol{c}_\ell \boldsymbol{c}_\ell^*, \boldsymbol{M} \rangle + \tilde{v}_\ell \langle \boldsymbol{b}_\ell \boldsymbol{b}_\ell^*, \boldsymbol{H} \rangle$. For a set $\mathcal{Q} \subset (\mathcal{H}^{k\times k}, \mathcal{H}^{n\times n})$, the Rademacher complexity of the gradients $\nabla f_\ell = (\frac{\partial f_\ell}{\partial \boldsymbol{H}}, \frac{\partial f_\ell}{\partial \boldsymbol{M}}) = (\tilde{v}_\ell \boldsymbol{b}_\ell \boldsymbol{b}_\ell^*, \tilde{u}_\ell \boldsymbol{c}_\ell \boldsymbol{c}_\ell^*)$ is defined as

$$\mathfrak{C}(\mathcal{Q}) := \mathbb{E} \sup_{(\boldsymbol{H}, \boldsymbol{M}) \in \mathcal{Q}} \frac{1}{\sqrt{m}} \sum_{\ell=1}^{m} \varepsilon_\ell \left\langle \nabla f_\ell, \frac{(\boldsymbol{H}, \boldsymbol{M})}{\|(\boldsymbol{H}, \boldsymbol{M})\|_F} \right\rangle, \tag{12}$$

where $\varepsilon_\ell, \ \ell = 1, 2, 3, \ldots, m$ are iid Rademacher random variables independent of everything else in the expression. For a convex set $\mathcal{Q}$, $\mathfrak{C}(\mathcal{Q})$ is a measure of the width of $\mathcal{Q}$ around origin interms of the gradients $\nabla f_\ell, \ \ell = 1, 2, 3, \ldots, m$. For example, random choice of gradient might yield little overlap with a structured set $\mathcal{Q}$ leading to a smaller complexity $\mathcal{Q}$.

Our result also depends on a probability $p_\tau(\mathcal{Q})$ and a positive parameter $\tau$ defined as

$$p_\tau(\mathcal{Q}) := \inf_{(\boldsymbol{H}, \boldsymbol{M}) \in \mathcal{Q}} \mathbb{P}\big( \langle \nabla f, (\boldsymbol{H}, \boldsymbol{M}) \rangle \ge \tau \|(\boldsymbol{H}, \boldsymbol{M})\|_F \big). \tag{13}$$

The probability $p_\tau(\mathcal{Q})$ quantifies visibility of the set $\mathcal{Q}$ through the gradient vectors $\nabla f$. A small value of $\tau$ and $p_\tau(\mathcal{Q})$ means that the set $\mathcal{Q}$ mainly remains invisible through the lenses of $\nabla f_\ell, \ell = 1, 2, 3, \ldots, m$. This can be appreciated just by noting that $p_\tau(\mathcal{Q})$ depends on the correlation of the elements of $\mathcal{Q}$ with the gradient vectors $\nabla f_\ell$.

Following lemma shows that the minimizer of the linear program (9) almost always resides in the desired set $(\tilde{\boldsymbol{H}}, \tilde{\boldsymbol{M}}) \oplus \mathcal{S}$ for a sufficiently large $m$ quantified interms of $\mathfrak{C}(\mathcal{Q})$, $p_\tau(\mathcal{Q})$, and $\tau$.

**Lemma 3.** *Consider the optimization program in (9) and $\mathcal{Q}$, characterized in (11), be the set of descent directions for which $\mathfrak{C}(\mathcal{Q})$, and $p_\tau(\mathcal{Q})$ can be determined using (12) and (13). Choose*

$$m \ge \left( \frac{2\mathfrak{C}(\mathcal{Q}) + t\tau}{\tau p_{\tau(\mathcal{Q})}} \right)^2$$

*for any $t > 0$. Then the minimizer $(\widehat{\boldsymbol{H}}, \widehat{\boldsymbol{M}})$ of (9) lies in the set $(\tilde{\boldsymbol{H}}, \tilde{\boldsymbol{M}}) \oplus \mathcal{S}$ with probability at least $1 - \mathrm{e}^{-2mt^2}$.*

Proof of this lemma is based on small ball method developed in Koltchinskii and Mendelson [2015], Mendelson [2014] and further studied in Lecué et al. [2018], Lecué and Mendelson [2017]. The proof is mainly repeated using the argument in Bahmani and Romberg [2017b], and is provided in the supplementary material for completeness.

With Lemma 3 in place, an application of Lemma 2 and the discussion after it proves that for choice of $m$ outlined in Lemma 3, $(\tilde{\boldsymbol{H}}, \tilde{\boldsymbol{M}})$ is the unique minimizer of (5). The last missing piece in the proof of Theorem 1 is the computation of the Rademacher complexity $\mathfrak{C}(\mathcal{Q})$, and $p_\tau(\mathcal{Q})$ for the $\mathcal{Q}$.

## 2.1 Rademacher Complexity

We begin with evaluation of the complexity $\mathfrak{C}(\mathcal{Q})$

$$\mathfrak{C}(\mathcal{Q}) := \mathbb{E} \sup_{(\delta\boldsymbol{H}, \delta\boldsymbol{M}) \in \mathcal{Q}} \frac{1}{\sqrt{m}} \sum_{\ell=1}^{m} \varepsilon_\ell \Big\langle \nabla f_\ell, \tfrac{(\delta\boldsymbol{H}, \delta\boldsymbol{M})}{\|(\delta\boldsymbol{H}, \delta\boldsymbol{M})\|_F} \Big\rangle$$

Splitting $(\delta\boldsymbol{H}, \delta\boldsymbol{M})$ between $(T_{\tilde{h}}, T_{\tilde{m}})$, and $(T_{\tilde{h}}^\perp, T_{\tilde{m}}^\perp)$, and using Holder's inequalities, we obtain

$$\mathfrak{C}(\mathcal{Q}) \le \mathbb{E}\Big\| \frac{1}{\sqrt{m}} \sum_{\ell=1}^{m} \varepsilon_\ell (\tilde{v}_\ell \mathcal{P}_{T_{\tilde{h}}}(\boldsymbol{b}_\ell \boldsymbol{b}_\ell^*), \tilde{u}_\ell \mathcal{P}_{T_{\tilde{m}}}(\boldsymbol{c}_\ell \boldsymbol{c}_\ell^*)) \Big\|_F \cdot \sup_{(\delta\boldsymbol{H}, \delta\boldsymbol{M}) \in \mathcal{Q}} \Big\| \tfrac{(\delta\boldsymbol{H}_{T_{\tilde{h}}}, \delta\boldsymbol{M}_{T_{\tilde{m}}})}{\|(\delta\boldsymbol{H}, \delta\boldsymbol{M})\|_F} \Big\|_F$$

$$+ \mathbb{E}\Big\| \frac{1}{\sqrt{m}} \sum_{\ell=1}^{m} \varepsilon_\ell (\tilde{v}_\ell \boldsymbol{b}_\ell \boldsymbol{b}_\ell^*, \tilde{u}_\ell \boldsymbol{c}_\ell \boldsymbol{c}_\ell^*) \Big\| \cdot \sup_{(\delta\boldsymbol{H}, \delta\boldsymbol{M}) \in \mathcal{Q}} \Big\| \tfrac{(\delta\boldsymbol{H}_{T_{\tilde{h}}^\perp}, \delta\boldsymbol{M}_{T_{\tilde{m}}^\perp})}{\|(\delta\boldsymbol{H}, \delta\boldsymbol{M})\|_F} \Big\|_*$$

On the set $\mathcal{Q}$, defined in (11), we have

$$\Big\| \tfrac{(\delta\boldsymbol{H}_{T_{\tilde{h}}^\perp}, \delta\boldsymbol{M}_{T_{\tilde{m}}^\perp})}{\|(\delta\boldsymbol{H}, \delta\boldsymbol{M})\|_F} \Big\|_* \le \Big\| \tfrac{(\delta\boldsymbol{H}_{T_{\tilde{h}}}, \delta\boldsymbol{M}_{T_{\tilde{m}}})}{\|(\delta\boldsymbol{H}, \delta\boldsymbol{M})\|_F} \Big\|_F \le 1.$$

Using Jensen's inequality, the first expectation simply becomes

$$\mathbb{E}\Big\| \frac{1}{\sqrt{m}} \sum_{\ell=1}^{m} \varepsilon_\ell (\tilde{v}_\ell \mathcal{P}_{T_{\tilde{h}}}(\boldsymbol{b}_\ell \boldsymbol{b}_\ell^*), \tilde{u}_\ell \mathcal{P}_{T_{\tilde{m}}}(\boldsymbol{c}_\ell \boldsymbol{c}_\ell^*)) \Big\|_F \le \sqrt{\frac{1}{m} \mathbb{E}\Big\| \sum_{\ell=1}^{m} \varepsilon_\ell (\tilde{v}_\ell \mathcal{P}_{T_{\tilde{h}}}(\boldsymbol{b}_\ell \boldsymbol{b}_\ell^*), \tilde{u}_\ell \mathcal{P}_{T_{\tilde{m}}}(\boldsymbol{c}_\ell \boldsymbol{c}_\ell^*)) \Big\|_F^2}$$

$$= \sqrt{\frac{1}{m} \sum_{\ell=1}^{m} \mathbb{E}\Big( \|\tilde{v}_\ell \mathcal{P}_{T_{\tilde{h}}}(\boldsymbol{b}_\ell \boldsymbol{b}_\ell^*)\|_F^2 + \|\tilde{u}_\ell \mathcal{P}_{T_{\tilde{m}}}(\boldsymbol{c}_\ell \boldsymbol{c}_\ell^*)\|_F^2 \Big)},$$

where the last equality follows by going through with the expectation over $\varepsilon_\ell$'s. Recall from the definition of the projection operator that $\mathcal{P}_{T_{\tilde{h}}}(\boldsymbol{b}_\ell \boldsymbol{b}_\ell^*) := \frac{\tilde{h} \tilde{h}^*}{\|\tilde{h}\|_2^2} \boldsymbol{b}_\ell \boldsymbol{b}_\ell^* + \boldsymbol{b}_\ell \boldsymbol{b}_\ell^* \frac{\tilde{h} \tilde{h}^*}{\|\tilde{h}\|_2^2} - \frac{\tilde{h} \tilde{h}^*}{\|\tilde{h}\|_2^2} \boldsymbol{b}_\ell \boldsymbol{b}_\ell^* \frac{\tilde{h} \tilde{h}^*}{\|\tilde{h}\|_2^2}$, and $\tilde{v}_\ell = |\boldsymbol{c}_\ell^* \tilde{m}|^2$. It can be easily verifies that $\|\mathcal{P}_{T_{\tilde{h}}}(\boldsymbol{b}_\ell \boldsymbol{b}_\ell^*)\|_F^2 = 2\frac{|\boldsymbol{b}_\ell^* \tilde{h}|^2}{\|\tilde{h}\|_2^2} \|\boldsymbol{b}_\ell\|_2^2 - \frac{|\boldsymbol{b}_\ell^* \tilde{h}|^4}{\|\tilde{h}\|_2^4}$, and, therefore,

$$\mathbb{E}\|\tilde{v}_\ell \mathcal{P}_{T_{\tilde{h}}}(\boldsymbol{b}_\ell \boldsymbol{b}_\ell^*)\|_F^2 \le \mathbb{E}|\boldsymbol{c}_\ell^* \tilde{m}|_2^4 \cdot \mathbb{E}\left( 2\frac{|\boldsymbol{b}_\ell^* \tilde{h}|^2}{\|\tilde{h}\|_2^2} \|\boldsymbol{b}_\ell\|_2^2 - \frac{|\boldsymbol{b}_\ell^* \tilde{h}|^4}{\|\tilde{h}\|_2^4} \right) \le 3\|\tilde{m}\|_2^4 \,(6k - 3),$$

where we used a simple calculation involving fourth moments of Gaussians $\mathbb{E}|\boldsymbol{b}_\ell^* \tilde{h}|^2 \|\boldsymbol{b}_\ell\|_2^2 = 3k\|\tilde{h}\|_2^2$. In an exactly similar manner, we can also show that $\|\tilde{u}_\ell \mathcal{P}_{T_{\tilde{m}}}(\boldsymbol{c}_\ell \boldsymbol{c}_\ell^*)\|_F^2 \le 3\|\tilde{h}\|_2^4 (6n - 3)$. Putting these together gives us

$$\mathbb{E}\Big\| \frac{1}{\sqrt{m}} \sum_{\ell=1}^{m} \varepsilon_\ell (\tilde{v}_\ell \mathcal{P}_{T_{\tilde{h}}}(\boldsymbol{b}_\ell \boldsymbol{b}_\ell^*), \tilde{u}_\ell \mathcal{P}_{T_{\tilde{m}}}(\boldsymbol{c}_\ell \boldsymbol{c}_\ell^*)) \Big\|_F \le 5 \max(\|\tilde{h}\|_2^2, \|\tilde{m}\|_2^2)\sqrt{k + n}.$$

Moreover,

$$\mathbb{E}\Big\| \frac{1}{\sqrt{m}} \sum_{\ell=1}^{m} \varepsilon_\ell (\tilde{v}_\ell \boldsymbol{b}_\ell \boldsymbol{b}_\ell^*, \tilde{u}_\ell \boldsymbol{c}_\ell \boldsymbol{c}_\ell^*) \Big\| \le \mathbb{E} \max_\ell (\tilde{u}_\ell, \tilde{v}_\ell) \cdot \mathbb{E}\Big\| \frac{1}{\sqrt{m}} \sum_{\ell=1}^{m} \varepsilon_\ell (\boldsymbol{b}_\ell \boldsymbol{b}_\ell^*, \boldsymbol{c}_\ell \boldsymbol{c}_\ell^*) \Big\|$$

Standard net arguments; see, for example, Sec. 5.4.1 of Eldar and Kutyniok [2012] show that

$$\mathbb{P}\left( \Big\| \frac{1}{\sqrt{m}} \sum_{\ell=1}^{m} \varepsilon_\ell (\boldsymbol{b}_\ell \boldsymbol{b}_\ell^*, \boldsymbol{c}_\ell \boldsymbol{c}_\ell^*) \Big\| \ge c\sqrt{k + n} \right) \le e^{-cm}, \text{ provided that } m \ge c(k + n).$$

This directly implies that $\mathbb{E}\Big\| \frac{1}{\sqrt{m}} \sum_{\ell=1}^{m} \varepsilon_\ell (\boldsymbol{b}_\ell \boldsymbol{b}_\ell^*, \boldsymbol{c}_\ell \boldsymbol{c}_\ell^*) \Big\| \le c\sqrt{k + n}$. The random variables $u_\ell$ and $v_\ell$ being sub-exponential have Orlicz-1 norms bounded by $c \max(\|\tilde{h}\|_2^2, \|\tilde{m}\|_2^2)$. Using standard results, such as Lemma 3 in van de Geer and Lederer [2013], we then have $\mathbb{E} \max_\ell(u_\ell, v_\ell) \le c \log m$. Putting these together yields

$$\mathbb{E}\Big\| \frac{1}{\sqrt{m}} \sum_{\ell=1}^{m} \varepsilon_\ell (\tilde{v}_\ell \boldsymbol{b}_\ell \boldsymbol{b}_\ell^*, \tilde{u}_\ell \boldsymbol{c}_\ell \boldsymbol{c}_\ell^*) \Big\| \le c \max(\|\tilde{h}\|_2^2, \|\tilde{m}\|_2^2)\sqrt{(k + n) \log^2 m}. \tag{14}$$

We have all the ingredients for the final bound on $\mathfrak{C}(\mathcal{Q})$ stated below

$$\mathfrak{C}(\mathcal{Q}) \le c \max(\|\tilde{h}\|_2^2, \|\tilde{m}\|_2^2)\sqrt{(k + n) \log^2 m}. \tag{15}$$

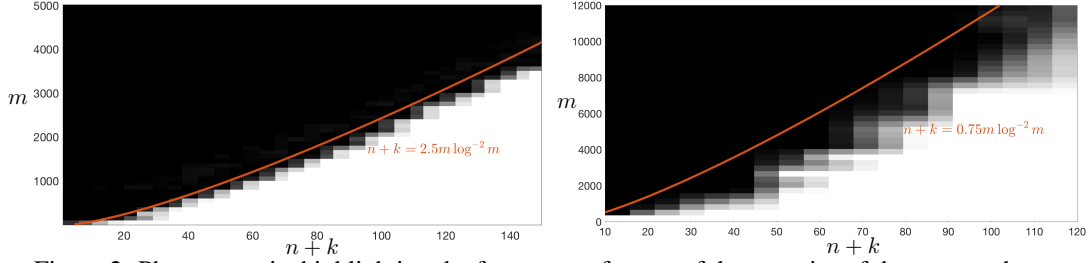

Figure 2: Phase portraits highlighting the frequency of successful recoveries of the proposed convex program for random and deterministic channel subspaces (see the text for the experiment details)

## 2.2 Probability $p_\tau(\mathcal{Q})$

The calculation for the probability $p_\tau(\mathcal{Q})$, and the positive parameter $\tau$ are given in Supplementary material due to limitation of space. We find that

$$p_\tau(\mathcal{Q}) \geq c > 0, \text{ and } \tau = c \max(\|\tilde{\boldsymbol{h}}\|_2^2, \|\tilde{\boldsymbol{m}}\|_2^2). \tag{16}$$

The complexity estimate in (15), value of $\tau$ computed above, and $p_\tau(\mathcal{Q})$ stated in (16) together with an application of Lemma 3 prove Theorem 1.

## 3 Convex Implementation and Phase Transition

To implement the semi-definite convex program (5), we propose a numerical scheme based on the alternating direction method of multipliers (ADMM). Due to the space limit, the technical details of the algorithm are moved to Section 4 of the supplementary note.

To illustrate the perfect recovery region, in Figure 2 we present the phase portrait associated with the proposed convex framework. To obtain the diagram on the left panel, for each fixed value of $m$, we run the algorithm for 100 different combinations of $n$ and $k$, each time using a different set of Gaussian matrices $\boldsymbol{B}$ and $\boldsymbol{C}$. If the algorithm converges to a sufficiently close neighborhood of the ground-truth solution (a distance less than 1% of the solution's $\ell_2$ norm), we label the experiment as successful. Figure 2 shows the collected success frequencies, where solid black corresponds to 100% success and solid white corresponds to 0% success. For an empirically selected constant $c$, the success region almost perfectly stands on the left side of the line $n + k = cm \log^{-2} m$.

While the analysis in this paper is specifically focused on the Gaussian subspace embeddings for $\boldsymbol{w}$ and $\boldsymbol{x}$, on the right panel of Figure 2 we have plotted the phase diagram for the case that $\boldsymbol{B}$ is deterministic and a subset of the columns of identity matrix (equispaced sampling of the columns), and $\boldsymbol{C}$ is Gaussian as before. This importantly hints that the convex framework is applicable to more realistic deterministic subspace models.

### Acknowledgments

PH acknowledges support from NSF DMS 1464525.

## Footnotes

[1]For a point $\boldsymbol{x}$, and a set $\mathcal{S}$, the notation $\boldsymbol{x} \oplus \mathcal{S}$ denotes a set of points $\boldsymbol{x} + \boldsymbol{s}_i$ for every $\boldsymbol{s}_i \in \mathcal{S}$.

[2] For brevity, we will often drop the dependence on $\boldsymbol{H}$, and $\boldsymbol{M}$ in the notation $f_\ell(\boldsymbol{H}, \boldsymbol{M})$

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
