[Supplementary Material]

# Supplementary Note:
# Blind Deconvolutional Phase Retrieval via Convex Programming

**Abstract**

The material presented in this document is supplementary to the manuscript submitted to NIPS 2018. The document contains extended discussions, additional technical proofs and details of the convex program implementation.

## 1  Visible Light Communication

As discussed in the body of the paper, an important application domain where blind deconvolution from phaseless Fourier measurements arises is the visible light communication (VLC). A stylized VLC setup is shown in Figure 1. A message $\boldsymbol{m} \in \mathbb{R}^n$ is to be transmitted using visible light. The message is first coded by multiplying it with a tall coding matrix $\boldsymbol{C} \in \mathbb{R}^{m \times n}$ and the resultant information $\boldsymbol{x} = \boldsymbol{C}\boldsymbol{m}$ is modulated on a light wave. The light wave propagates through an unknown media. This propagation can be modeled as a convolution $\boldsymbol{x} \circledast \boldsymbol{w}$ of the information signal $\boldsymbol{x}$ with unknown channel $\boldsymbol{w} \in \mathbb{R}^m$. The vector $\boldsymbol{w}$ contains channel taps, and frequently in realistic applications has only few significant taps. In this case, one can model

$$\boldsymbol{w} \approx \boldsymbol{B}\boldsymbol{h},$$

where $\boldsymbol{h} \in \mathbb{R}^k$ is a short $(k \ll m)$ vector, and $\boldsymbol{B} \in \mathbb{R}^{m \times k}$ in this case is a subset of the columns of an identity matrix. Generally, the multipath channels are well modeled with non-zero taps in top locations of $\boldsymbol{w}$. In that case, $\boldsymbol{B}$ is exactly known to be top few columns of the identity matrix.

In visible light communication, there is always a difficulty associated with measuring phase information in the received light. Figure 1 shows a setup, where we measure the phaseless Fourier transform (light through the lens) of this signal. The measurements are therefore

$$\boldsymbol{y} = |\boldsymbol{F}(\boldsymbol{C}\boldsymbol{m} \circledast \boldsymbol{B}\boldsymbol{h})|$$

and one wants to recover $\boldsymbol{m}$, and $\boldsymbol{h}$ given the knowledge of $\boldsymbol{B}$, and the coding matrix $\boldsymbol{C}$. Since we chose $\boldsymbol{C}$ to be random Gaussian, and $\boldsymbol{B}$ is the columns of identity. As mentioned at the

Figure 1: Visible light communication optical setup; the media block normally consists of phosphor, filter and a linear polarizer. The lens takes the Fourier transform of the light and one can only measure the intensity only measurements of this transformed light source signal.

end of the numerics section that with this subspace model, we obtain similar recovery results as one would have for both $\boldsymbol{B}$, and $\boldsymbol{C}$ being random Gaussians. The proposed convex program solves this difficult inverse problem and recovers the true solution with these subspace models.

## 2    Proof of Lemma 3

The proof is based on small ball method developed in [Koltchinskii and Mendelson(2015), Mendelson(2014)] and further studied in [Lecué et al.(2018)Lecué, Mendelson, et al.] and [Lecué and Mendelson(2017)]. The proof is mainly repeated using a similar line of argument as in [Bahmani and Romberg(2017)], and is provided here for completeness.

Rest of the proof now concerns showing that $(\tilde{\boldsymbol{H}}, \tilde{\boldsymbol{M}})$ is the unique solution to the linearly constrained optimization program (9). Define one sided loss function:

$$\mathcal{L}(\boldsymbol{H}, \boldsymbol{M}) := \sum_{\ell=1}^{m} \left( 2y_\ell^2 - \tfrac{1}{m}\langle \boldsymbol{b}_\ell \boldsymbol{b}_\ell^*, \boldsymbol{H}\rangle \langle \boldsymbol{c}_\ell \boldsymbol{c}_\ell^*, \tilde{\boldsymbol{M}}\rangle - \langle \boldsymbol{b}_\ell \boldsymbol{b}_\ell^*, \tilde{\boldsymbol{H}}\rangle \langle \boldsymbol{c}_\ell \boldsymbol{c}_\ell^*, \boldsymbol{M}\rangle \right)_+, \qquad \text{(S.1)}$$

where $(\cdot)_+$ denotes the positive side. Using this definition, we rewrite (9) compactly as

$$\begin{aligned} \text{minimize} \quad & \|\boldsymbol{H}\|_* + \|\boldsymbol{M}\|_* \\ \text{subject to} \quad & \mathcal{L}(\boldsymbol{H}, \boldsymbol{M}) \le 0. \end{aligned} \qquad \text{(S.2)}$$

The goal of the proof is to show that all descent direction $(\delta \boldsymbol{H}, \delta \boldsymbol{M}) \in \mathcal{Q}$ that also obey the constraint set have a small $\ell_2$ norm. Since $(\delta \boldsymbol{H}, \delta \boldsymbol{M})$ is a feasible perturbation from the proposed optimal $(\tilde{\boldsymbol{H}}, \tilde{\boldsymbol{M}})$, we have from the constraints above that

$$\mathcal{L}\left(\tilde{\boldsymbol{H}} + \delta \boldsymbol{H}, \tilde{\boldsymbol{M}} + \delta \boldsymbol{M}\right) \le 0 \qquad \text{(S.3)}$$

We begin by expanding the loss function $\text{Loss}(\tilde{\boldsymbol{H}} + \delta\boldsymbol{H}, \tilde{\boldsymbol{M}} + \delta\boldsymbol{M})$ below

$$
\begin{aligned}
\mathcal{L}(\tilde{\boldsymbol{H}} &+ \delta\boldsymbol{H}, \tilde{\boldsymbol{M}} + \delta\boldsymbol{M}) \\
&= \sum_{\ell=1}^{m} \left( 2y_\ell^2 - \frac{1}{m}\big( \langle \boldsymbol{b}_\ell \boldsymbol{b}_\ell^*, \tilde{\boldsymbol{H}} + \delta\boldsymbol{H}\rangle\langle \boldsymbol{c}_\ell \boldsymbol{c}_\ell^*, \tilde{\boldsymbol{M}}\rangle + \langle \boldsymbol{b}_\ell \boldsymbol{b}_\ell^*, \tilde{\boldsymbol{H}}\rangle\langle \boldsymbol{c}_\ell \boldsymbol{c}_\ell^*, \tilde{\boldsymbol{M}} + \delta\boldsymbol{M}\rangle \big) \right)_+ \\
&= \frac{1}{m}\sum_{\ell=1}^{m} \Big( \big( \langle \boldsymbol{b}_\ell \boldsymbol{b}_\ell^*, \tilde{\boldsymbol{H}}\rangle\langle \boldsymbol{c}_\ell \boldsymbol{c}_\ell^*, \tilde{\boldsymbol{M}}\rangle + \langle \boldsymbol{b}_\ell \boldsymbol{b}_\ell^*, \tilde{\boldsymbol{H}}\rangle\langle \boldsymbol{c}_\ell \boldsymbol{c}_\ell^*, \tilde{\boldsymbol{M}}\rangle \big) - \\
&\qquad\qquad \big( \langle \boldsymbol{b}_\ell \boldsymbol{b}_\ell^*, \tilde{\boldsymbol{H}} + \delta\boldsymbol{H}\rangle\langle \boldsymbol{c}_\ell \boldsymbol{c}_\ell^*, \tilde{\boldsymbol{M}}\rangle + \langle \boldsymbol{b}_\ell \boldsymbol{b}_\ell^*, \tilde{\boldsymbol{H}}\rangle\langle \boldsymbol{c}_\ell \boldsymbol{c}_\ell^*, \tilde{\boldsymbol{M}} + \delta\boldsymbol{M}\rangle \big) \Big)_+ \\
&= \frac{1}{m}\sum_{\ell=1}^{m} \Big( -\langle \boldsymbol{b}_\ell \boldsymbol{b}_\ell^*, \delta\boldsymbol{H}\rangle\langle \boldsymbol{c}_\ell \boldsymbol{c}_\ell^*, \tilde{\boldsymbol{M}}\rangle - \langle \boldsymbol{b}_\ell \boldsymbol{b}_\ell^*, \tilde{\boldsymbol{H}}\rangle\langle \boldsymbol{c}_\ell \boldsymbol{c}_\ell^*, \delta\boldsymbol{M}\rangle \Big)_+ \\
&\geq \frac{1}{m}\sum_{\ell=1}^{m} \big( (-\langle \nabla f_\ell, (\delta\boldsymbol{H}, \delta\boldsymbol{M})\rangle) \big)_+ . \qquad\qquad (\text{S.4})
\end{aligned}
$$

where the last equality follows from the using notation $\nabla f_\ell = (\tilde{v}_\ell \boldsymbol{b}_\ell \boldsymbol{b}_\ell^*, \tilde{u}_\ell \boldsymbol{c}_\ell \boldsymbol{c}_\ell^*)$ introduced earlier. Let $\psi_t(s) := (s)_+ - (s-t)_+$. Using the fact that $\psi_t(s) \leq (s)_+$, and that for every $\alpha, t \geq 0$, and $s \in \mathbb{R}$, $\psi_{\alpha t}(s) = t\psi_\alpha(\frac{s}{t})$, we have

$$
\begin{aligned}
\frac{1}{m}\sum_{\ell=1}^{m} \big[ -\langle \nabla f_\ell, (\delta\boldsymbol{H}, \delta\boldsymbol{M})\rangle \big]_+ &\geq \frac{1}{m}\sum_{\ell=1}^{m} \psi_{\tau\|(\delta\boldsymbol{H},\delta\boldsymbol{M})\|_F}\Big( -\langle \nabla f_\ell, (\delta\boldsymbol{H}, \delta\boldsymbol{M})\rangle \Big) \\
&= \|(\delta\boldsymbol{H}, \delta\boldsymbol{M})\|_F \cdot \frac{1}{m}\sum_{\ell=1}^{m} \psi_\tau\Big( -\big\langle \nabla f_\ell, \tfrac{(\delta\boldsymbol{H},\delta\boldsymbol{M})}{\|(\delta\boldsymbol{H},\delta\boldsymbol{M})\|_F}\big\rangle \Big) \\
&= \|(\delta\boldsymbol{H}, \delta\boldsymbol{M})\|_F \left[ \frac{1}{m}\sum_{\ell=1}^{m} \mathbb{E}\psi_\tau\Big( -\big\langle \nabla f_\ell, \tfrac{(\delta\boldsymbol{H},\delta\boldsymbol{M})}{\|(\delta\boldsymbol{H},\delta\boldsymbol{M})\|_F}\big\rangle \Big) \right. \\
&\quad \left. - \frac{1}{m}\sum_{\ell=1}^{m} \left[ \mathbb{E}\psi_\tau\Big( -\big\langle \nabla f_\ell, \tfrac{(\delta\boldsymbol{H},\delta\boldsymbol{M})}{\|(\delta\boldsymbol{H},\delta\boldsymbol{M})\|_F}\big\rangle \Big) - \psi_\tau\Big( -\big\langle \nabla f_\ell, \tfrac{(\delta\boldsymbol{H},\delta\boldsymbol{M})}{\|(\delta\boldsymbol{H},\delta\boldsymbol{M})\|_F}\big\rangle \Big) \right] \right]. \qquad (\text{S.5})
\end{aligned}
$$

Define a centered random process $\mathcal{R}(\boldsymbol{B}, \boldsymbol{C})$ as follows

$$
\mathcal{R}(\boldsymbol{B}, \boldsymbol{C}) := \sup_{(\delta\boldsymbol{H},\delta\boldsymbol{M})\in\mathcal{Q}} \frac{1}{m}\sum_{\ell=1}^{m} \left[ \mathbb{E}\psi_\tau\Big( -\big\langle \nabla f_\ell, \tfrac{(\delta\boldsymbol{H},\delta\boldsymbol{M})}{\|(\delta\boldsymbol{H},\delta\boldsymbol{M})\|_F}\big\rangle \Big) - \psi_\tau\Big( -\big\langle \nabla f_\ell, \tfrac{(\delta\boldsymbol{H},\delta\boldsymbol{M})}{\|(\delta\boldsymbol{H},\delta\boldsymbol{M})\|_F}\big\rangle \Big) \right]
$$

and an application of bounded difference inequality [McDiarmid(1989)] yields that $\mathcal{R}(\boldsymbol{B}, \boldsymbol{C}) \leq \mathbb{E}\mathcal{R}(\boldsymbol{B}, \boldsymbol{C}) + t\tau/\sqrt{m}$ with probability at least $1 - e^{-2mt^2}$. It remains to evaluate $\mathbb{E}\mathcal{R}(\boldsymbol{B}, \boldsymbol{C})$, which after using a simple symmetrization inequality [van der Vaart and Wellner(1997)] yields

$$
\mathbb{E}\mathcal{R}(\boldsymbol{B}, \boldsymbol{C}) \leq 2\mathbb{E} \sup_{(\delta\boldsymbol{H},\delta\boldsymbol{M})\in\mathcal{Q}} \frac{1}{m}\sum_{\ell=1}^{m} \varepsilon_\ell \psi_\tau\Big( -\big\langle \nabla f_\ell, \tfrac{(\delta\boldsymbol{H},\delta\boldsymbol{M})}{\|(\delta\boldsymbol{H},\delta\boldsymbol{M})\|_F}\big\rangle \Big), \qquad (\text{S.6})
$$

where $\varepsilon_1, \varepsilon_2, \ldots, \varepsilon_m$ are independent Rademacher random variables. Using the fact that $\psi_t(0) = 0$, and $\psi_t(s)$ is a contraction: $|\psi_t(\alpha_1) - \psi_t(\alpha_2)| \leq |\alpha_1 - \alpha_2|$ for all $\alpha_1, \alpha_2 \in \mathbb{R}$, we have from the Rademacher contraction inequality Theorem 4.12 in [Ledoux and Talagrand(2013)] that

$$\mathbb{E} \sup_{(\delta \boldsymbol{H}, \delta \boldsymbol{M}) \in \mathcal{Q}} \frac{1}{m} \sum_{\ell=1}^{m} \varepsilon_\ell \psi_\tau \left( - \left\langle \nabla f_\ell, \frac{(\delta \boldsymbol{H}, \delta \boldsymbol{M})}{\|(\delta \boldsymbol{H}, \delta \boldsymbol{M})\|_F} \right\rangle \right) \leq \mathbb{E} \sup_{(\delta \boldsymbol{H}, \delta \boldsymbol{M}) \in \mathcal{Q}} \frac{1}{m} \sum_{\ell=1}^{m} -\varepsilon_\ell \left\langle \nabla f_\ell, \frac{(\delta \boldsymbol{H}, \delta \boldsymbol{M})}{\|(\delta \boldsymbol{H}, \delta \boldsymbol{M})\|_F} \right\rangle$$

$$= \mathbb{E} \sup_{(\delta \boldsymbol{H}, \delta \boldsymbol{M}) \in \mathcal{Q}} \frac{1}{m} \sum_{\ell=1}^{m} \varepsilon_\ell \left\langle \nabla f_\ell, \frac{(\delta \boldsymbol{H}, \delta \boldsymbol{M})}{\|(\delta \boldsymbol{H}, \delta \boldsymbol{M})\|_F} \right\rangle, \tag{S.7}$$

where the last equality is the result of the fact that a global sign change of a sequence of Rademacher random variables does not change their distribution. In addition, using the facts that $t\mathbf{1}(s \geq t) \leq \psi_t(s)$, and that random vectors $\nabla f_1, \nabla f_2, \ldots, \nabla f_m$ are identically distributed and the distribution is symmetric, it follows

$$\tau \mathbb{P}\left( \left\langle \nabla f_\ell, \frac{(\delta \boldsymbol{H}, \delta \boldsymbol{M})}{\|(\delta \boldsymbol{H}, \delta \boldsymbol{M})\|_F} \right\rangle \geq \tau \right) = \tau \mathbb{E}\left( \mathbf{1}\left[ \left\langle \nabla f_\ell, \frac{(\delta \boldsymbol{H}, \delta \boldsymbol{M})}{\|(\delta \boldsymbol{H}, \delta \boldsymbol{M})\|_F} \right\rangle \geq \tau \right] \right)$$

$$\leq \mathbb{E} \psi_\tau \left( \left\langle \nabla f_\ell, \frac{(\delta \boldsymbol{H}, \delta \boldsymbol{M})}{\|(\delta \boldsymbol{H}, \delta \boldsymbol{M})\|_F} \right\rangle \right). \tag{S.8}$$

Plugging (S.8), and (S.7) in (S.5), we have

$$\frac{1}{m} \sum_{\ell=1}^{m} \left( - \left\langle \nabla f_\ell, \frac{(\delta \boldsymbol{H}, \delta \boldsymbol{M})}{\|(\delta \boldsymbol{H}, \delta \boldsymbol{M})\|_F} \right\rangle \right)_+ \geq \tau \|(\delta \boldsymbol{H}, \delta \boldsymbol{M})\|_F \cdot \mathbb{P}\left( \left\langle \nabla f_\ell, \frac{(\delta \boldsymbol{H}, \delta \boldsymbol{M})}{\|(\delta \boldsymbol{H}, \delta \boldsymbol{M})\|_F} \right\rangle \geq \tau \right)$$

$$- 2\|(\delta \boldsymbol{H}, \delta \boldsymbol{M})\|_F \cdot \mathbb{E} \sup_{(\delta \boldsymbol{H}, \delta \boldsymbol{M}) \in \mathcal{Q}} \left[ \frac{1}{m} \sum_{\ell=1}^{m} \varepsilon_\ell \left\langle \nabla f_\ell, \frac{(\delta \boldsymbol{H}, \delta \boldsymbol{M})}{\|(\delta \boldsymbol{H}, \delta \boldsymbol{M})\|_F} \right\rangle \right] - 2\|(\delta \boldsymbol{H}, \delta \boldsymbol{M})\|_F \frac{t\tau}{\sqrt{m}}.$$

Combining this with (S.3) and (S.4), we obtain the final result

$$\|(\delta \boldsymbol{H}, \delta \boldsymbol{M})\|_F \left[ \tau \mathbb{P}\left( \left\langle \nabla f_\ell, \frac{(\delta \boldsymbol{H}, \delta \boldsymbol{M})}{\|(\delta \boldsymbol{H}, \delta \boldsymbol{M})\|_F} \right\rangle \geq \tau \right) - 2\mathbb{E} \sup_{(\delta \boldsymbol{H}, \delta \boldsymbol{M}) \in \mathcal{Q}} \frac{1}{m} \sum_{\ell=1}^{m} \varepsilon_\ell \left\langle \nabla f_\ell, \frac{(\delta \boldsymbol{H}, \delta \boldsymbol{M})}{\|(\delta \boldsymbol{H}, \delta \boldsymbol{M})\|_F} \right\rangle \right]$$

$$- 2\|(\delta \boldsymbol{H}, \delta \boldsymbol{M})\|_F \frac{t\tau}{\sqrt{m}} \leq 0.$$

Using the definitions in (12), and (13), we can write

$$\|(\delta \boldsymbol{H}, \delta \boldsymbol{M})\|_F \left( \tau p_\tau(\mathcal{Q}) - \frac{(2\mathfrak{C}(\mathcal{Q}) + t\tau)}{\sqrt{m}} \right) \leq 0.$$

It is clear that choosing $m \geq \left( \frac{2\mathfrak{C}(\mathcal{Q}) + t\tau}{\tau p_\tau(\mathcal{Q})} \right)^2$ implies

$$(\delta \boldsymbol{H}, \delta \boldsymbol{M}) = (\boldsymbol{0}, \boldsymbol{0}).$$

The proof is complete.

# 3 Probability $p_\tau(\mathcal{Q})$

In this section, we determine the probability $p_\tau(\mathcal{Q})$, and the positive parameter $\tau$ in (13) for the set $\mathcal{Q}$ in (11). For a point $(\delta\boldsymbol{H}, \delta\boldsymbol{M}) \in \mathcal{Q}$, and randomly chosen $\nabla f_\ell$, we have via Paley Zygmund inequality that

$$\mathbb{P}\Big(|\langle\nabla f_\ell, (\delta\boldsymbol{H}, \delta\boldsymbol{M})\rangle|^2 \geq \frac{1}{2}\mathbb{E}\,|\langle\nabla f_\ell, (\delta\boldsymbol{H}, \delta\boldsymbol{M})\rangle|^2\Big) \geq \frac{1}{4}\frac{\big(\mathbb{E}\,|\langle\nabla f_\ell, (\delta\boldsymbol{H}, \delta\boldsymbol{M})\rangle|^2\big)^2}{\mathbb{E}\,|\langle\nabla f_\ell, (\delta\boldsymbol{H}, \delta\boldsymbol{M})\rangle|^4}.$$

The particular choice of random gradient vectors we are using is $\nabla f_\ell = (\tilde{v}_\ell\boldsymbol{b}_\ell\boldsymbol{b}_\ell^*, \tilde{u}_\ell\boldsymbol{c}_\ell\boldsymbol{c}_\ell^*)$ giving us $\langle\nabla f_\ell, (\delta\boldsymbol{H}, \delta\boldsymbol{M})\rangle = \tilde{v}_\ell\langle\boldsymbol{b}_\ell\boldsymbol{b}_\ell^*, \delta\boldsymbol{H}\rangle + \tilde{u}_\ell\langle\boldsymbol{c}_\ell\boldsymbol{c}_\ell^*, \delta\boldsymbol{M}\rangle$. Since $\boldsymbol{b}_\ell$, and $\boldsymbol{c}_\ell$ are standard Gaussian vectors, using the equivalence of $L_p$-norms for Gaussians, we deduce that

$$\Big(\mathbb{E}\,|\tilde{v}_\ell\langle\boldsymbol{b}_\ell\boldsymbol{b}_\ell^*, \delta\boldsymbol{H}\rangle + \tilde{u}_\ell\langle\boldsymbol{c}_\ell\boldsymbol{c}_\ell^*, \delta\boldsymbol{M}\rangle|^4\Big)^{1/4} \leq c\Big(\mathbb{E}\,|\tilde{v}_\ell\langle\boldsymbol{b}_\ell\boldsymbol{b}_\ell^*, \delta\boldsymbol{H}\rangle + \tilde{u}_\ell\langle\boldsymbol{c}_\ell\boldsymbol{c}_\ell^*, \delta\boldsymbol{M}\rangle|^2\Big)^{1/2}.$$

Plugging last two inequalities in (13) reveals that

$$p_\tau(\mathcal{Q}) \geq c > 0 \tag{S.9}$$

for an absolute constant $c$. To compute $\tau$, we expand $\mathbb{E}\,|\langle\nabla f_\ell, (\delta\boldsymbol{H}, \delta\boldsymbol{M})\rangle|^2$ giving us

$$\begin{aligned}
\mathbb{E}\,|\tilde{v}_\ell\langle\boldsymbol{b}_\ell\boldsymbol{b}_\ell^*, \delta\boldsymbol{H}\rangle + \tilde{u}_\ell\langle\boldsymbol{c}_\ell\boldsymbol{c}_\ell^*, \delta\boldsymbol{M}\rangle|^2 &= 3\|\tilde{\boldsymbol{m}}\|_2^4(\langle\mathrm{diag}(\delta\boldsymbol{H}), \delta\boldsymbol{H}\rangle + 2\|\delta\boldsymbol{H}\|_F^2) \\
&\quad + 3\|\tilde{\boldsymbol{h}}\|_2^4(\langle\mathrm{diag}(\delta\boldsymbol{M}), \delta\boldsymbol{M}\rangle + 2\|\delta\boldsymbol{M}\|_F^2) + 2|\tilde{\boldsymbol{h}}^*\mathrm{diag}(\delta\boldsymbol{H})\tilde{\boldsymbol{h}} + 2\tilde{\boldsymbol{h}}^*\delta\boldsymbol{H}\tilde{\boldsymbol{h}}|^2, \tag{S.10}
\end{aligned}$$

where we have made use of multiple simple facts including that $\mathbb{E}|\tilde{u}_\ell|^2 = 3\|\tilde{\boldsymbol{h}}\|_2^4$, and similarly for $\tilde{v}_\ell$, and two identities: $\mathbb{E}|\boldsymbol{b}_\ell^*\tilde{\boldsymbol{h}}|^2\boldsymbol{b}_\ell^*\delta\boldsymbol{H}\boldsymbol{b}_\ell = \tilde{\boldsymbol{h}}^*\mathrm{diag}(\delta\boldsymbol{H})\tilde{\boldsymbol{h}} + 2\tilde{\boldsymbol{h}}^*\delta\boldsymbol{H}\tilde{\boldsymbol{h}}$, and $\mathbb{E}(\boldsymbol{b}_\ell^*\delta\boldsymbol{H}\boldsymbol{b}_\ell)\boldsymbol{b}_\ell\boldsymbol{b}_\ell^* = \mathrm{diag}(\delta\boldsymbol{H}) + 2(\delta\boldsymbol{H}) \implies \mathbb{E}|\boldsymbol{b}_\ell^*\delta\boldsymbol{H}\boldsymbol{b}_\ell|^2 = \langle\mathrm{diag}(\delta\boldsymbol{H}), \delta\boldsymbol{H}\rangle + 2\|\delta\boldsymbol{H}\|_F^2$. We also made use of the fact that $\mathcal{Q} \perp \mathcal{N}$ and therefore $\langle\tilde{\boldsymbol{H}}, \delta\boldsymbol{H}\rangle - \langle\tilde{\boldsymbol{M}}, \delta\boldsymbol{M}\rangle = 0$, or equivalently, $\tilde{\boldsymbol{h}}^*\delta\boldsymbol{H}\tilde{\boldsymbol{h}} = \tilde{\boldsymbol{m}}^*\delta\boldsymbol{M}\tilde{\boldsymbol{m}}$.

It is easy to conclude from (S.10) now that

$$\begin{aligned}
\mathbb{E}\,|\tilde{v}_\ell\langle\boldsymbol{b}_\ell\boldsymbol{b}_\ell^*, \delta\boldsymbol{H}\rangle + \tilde{u}_\ell\langle\boldsymbol{c}_\ell\boldsymbol{c}_\ell^*, \delta\boldsymbol{M}\rangle|^2 &\geq 6(\|\tilde{\boldsymbol{h}}\|_2^4\|\delta\boldsymbol{H}\|_F^2 + \|\tilde{\boldsymbol{m}}\|_2^4\|\delta\boldsymbol{M}\|_F^2) \\
&\geq c\min(\|\tilde{\boldsymbol{h}}\|_2^2, \|\tilde{\boldsymbol{m}}\|_2^2)(\|\delta\boldsymbol{H}\|_F^2 + \|\delta\boldsymbol{M}\|_F^2) = c\max(\|\tilde{\boldsymbol{h}}\|_2^2, \|\tilde{\boldsymbol{m}}\|_2^2)(\|\delta\boldsymbol{H}\|_F^2 + \|\delta\boldsymbol{M}\|_F^2),
\end{aligned}$$

where the last equality uses the fact that $\mathrm{Tr}(\tilde{\boldsymbol{H}}) = \mathrm{Tr}(\tilde{\boldsymbol{M}})$ from (8), which is equivalent to $\|\tilde{\boldsymbol{h}}\|_2^2 = \tilde{\boldsymbol{m}}\|_2^2$. This directly means, we can take $\tau = c\max(\|\tilde{\boldsymbol{h}}\|_2^2, \|\tilde{\boldsymbol{m}}\|_2^2)$, where $c$ is an absolute constant.

# 4    Implementing the Convex Program

In this section we take an alternating direction method of multipliers (ADMM) scheme to address (5), which takes the form

$$\underset{\boldsymbol{X}_1, \boldsymbol{X}_2}{\text{minimize}} \;\; \text{Tr}(\boldsymbol{X}_1) + \text{Tr}(\boldsymbol{X}_2) \tag{S.11}$$

$$\text{subject to} \;\; \langle \boldsymbol{a}_{1,\ell}\boldsymbol{a}_{1,\ell}{}^*, \boldsymbol{X}_1 \rangle \langle \boldsymbol{a}_{2,\ell}\boldsymbol{a}_{2,\ell}{}^*, \boldsymbol{X}_2 \rangle \geq \delta_\ell \geq 0,$$

$$\ell = 1, 2, \ldots, L,$$

$$\boldsymbol{X}_1 \succcurlyeq \boldsymbol{0}, \;\; \boldsymbol{X}_2 \succcurlyeq \boldsymbol{0}.$$

Note that for a complex matrix $\boldsymbol{X}$ being Hermitian is a requirement for being positive semidefinite. For a simpler notation we define the convex set

$$\mathcal{C} = \left\{ (\boldsymbol{u}, \boldsymbol{v}) \in \mathbb{R}^L \times \mathbb{R}^L : u_\ell v_\ell \geq \delta_\ell > 0, u_\ell \geq 0 \right\}. \tag{S.12}$$

In order to derive the ADMM scheme, after introducing new variables, program (S.11) can be written as

$$\underset{\{\boldsymbol{X}_i, \boldsymbol{Z}_i, \boldsymbol{u}_i\}_{i=1,2}}{\text{minimize}} \;\; \mathbb{I}_{\mathcal{C}}(\boldsymbol{u}_1, \boldsymbol{u}_2) + \sum_{j=1}^{2} \text{Tr}(\boldsymbol{X}_j) + \mathbb{I}_{+}(\boldsymbol{Z}_j) \tag{S.13}$$

$$\text{subject to} \quad u_{j,\ell} = \langle \boldsymbol{a}_{j,\ell}\boldsymbol{a}_{j,\ell}{}^*, \boldsymbol{X}_j \rangle, \;\; \ell = 1, 2, \ldots, L, \;\; j = 1, 2,$$

$$\boldsymbol{X}_j = \boldsymbol{Z}_j, \;\; j = 1, 2,$$

where the constraints are reflected in the indicator functions

$$\mathbb{I}_{\mathcal{C}}(\boldsymbol{u}, \boldsymbol{v}) = \left\{ \begin{array}{ll} 0 & (\boldsymbol{u}, \boldsymbol{v}) \in \mathcal{C} \\ +\infty & (\boldsymbol{u}, \boldsymbol{v}) \notin \mathcal{C} \end{array} \right., \quad \mathbb{I}_{+}(\boldsymbol{Z}) = \left\{ \begin{array}{ll} 0 & \boldsymbol{Z} \succeq \boldsymbol{0} \\ +\infty & \boldsymbol{Z} \not\succeq \boldsymbol{0} \end{array} \right..$$

Defining the dual matrices $\boldsymbol{P}_1, \boldsymbol{P}_2$ and the dual vectors $\boldsymbol{\alpha}_1, \boldsymbol{\alpha}_2 \in \mathbb{R}^L$, the augmented Lagrangian for (S.13) takes the form

$$L\left(\{\boldsymbol{X}_i, \boldsymbol{Z}_i, \boldsymbol{P}_i, \boldsymbol{u}_i, \boldsymbol{\alpha}_i\}_{i=1,2}\right) = \mathbb{I}_{\mathcal{C}}(\boldsymbol{u}_1, \boldsymbol{u}_2) + \sum_{j=1}^{2} \text{Tr}(\boldsymbol{X}_j) + \mathbb{I}_{+}(\boldsymbol{Z}_j)$$

$$+ \frac{\rho_1}{2} \sum_{j=1}^{2} \sum_{\ell=1}^{L} \left( u_{j,\ell} - \langle \boldsymbol{a}_{j,\ell}\boldsymbol{a}_{j,\ell}{}^*, \boldsymbol{X}_j \rangle + \alpha_{j,\ell} \right)^2$$

$$+ \frac{\rho_2}{2} \sum_{j=1}^{2} \|\boldsymbol{X}_j - \boldsymbol{Z}_j + \boldsymbol{P}_j\|_F^2. \tag{S.14}$$

In an ADMM scheme the update for each variable at the $k$-th iteration is performed by minimizing $L$ with respect to that variable, while fixing the other ones. More specifically, using the superscript $(k)$ to denote the iteration, for $j = 1, 2$ we have the primal updates

$$\boldsymbol{X}_j^{(k+1)} = \arg\min_{\boldsymbol{X}_j} \operatorname{Tr}(\boldsymbol{X}_j) + \frac{\rho_1}{2} \sum_{\ell=1}^{L} \left( \langle \boldsymbol{a}_{j,\ell}\boldsymbol{a}_{j,\ell}^*, \boldsymbol{X}_j \rangle - u_{j,\ell}^{(k)} - \alpha_{j,\ell}^{(k)} \right)^2 + \frac{\rho_2}{2} \left\| \boldsymbol{X}_j - \boldsymbol{Z}_j^{(k)} + \boldsymbol{P}_j^{(k)} \right\|_F^2,$$

$$\boldsymbol{Z}_j^{(k+1)} = \arg\min_{\boldsymbol{Z}_j} \frac{1}{2} \left\| \boldsymbol{Z}_j - \boldsymbol{X}_j^{(k+1)} - \boldsymbol{P}_j^{(k)} \right\|_F^2 + \mathbb{I}_+(\boldsymbol{Z}_j),$$

$$\left( \boldsymbol{u}_1^{(k+1)}, \boldsymbol{u}_2^{(k+1)} \right) = \arg\min_{\boldsymbol{u}_1, \boldsymbol{u}_2} \frac{1}{2} \sum_{j=1}^{2} \sum_{\ell=1}^{L} \left( u_{j,\ell} - \left\langle \boldsymbol{a}_{j,\ell}\boldsymbol{a}_{j,\ell}^*, \boldsymbol{X}_j^{(k+1)} \right\rangle + \alpha_{j,\ell}^{(k)} \right)^2 + \mathbb{I}_{\mathcal{C}}(\boldsymbol{u}_1, \boldsymbol{u}_2),$$

and the dual updates

$$\alpha_{j,\ell}^{(k+1)} = \alpha_{j,\ell}^{(k)} + u_{j,\ell}^{(k+1)} - \left\langle \boldsymbol{a}_{j,\ell}\boldsymbol{a}_{j,\ell}^*, \boldsymbol{X}_j^{(k+1)} \right\rangle$$

$$\boldsymbol{P}_j^{(k+1)} = \boldsymbol{P}_j^{(k)} + \boldsymbol{X}_j^{(k+1)} - \boldsymbol{Z}_j^{(k+1)}.$$

In the sequel we derive closed-form expressions for all the primal updates. To formulate the $\boldsymbol{X}$-update, taking the derivative of the objective with respect to $\boldsymbol{X}_j$ and setting it to zero yields

$$\boldsymbol{I} + \rho_1 \sum_{\ell=1}^{L} \left( \left\langle \boldsymbol{a}_{j,\ell}\boldsymbol{a}_{j,\ell}^*, \boldsymbol{X}_j^{(k+1)} \right\rangle - u_{j,\ell}^{(k)} - \alpha_{j,\ell}^{(k)} \right) \boldsymbol{a}_{j,\ell}\boldsymbol{a}_{j,\ell}^* + \rho_2 \left( \boldsymbol{X}_j^{(k+1)} - \boldsymbol{Z}_j^{(k)} + \boldsymbol{P}_j^{(k)} \right) = \boldsymbol{0},$$

which after vectorizing $\boldsymbol{X}_j^{(k+1)}$ yields

$$\operatorname{vec}\left( \boldsymbol{X}_j^{(k+1)} \right) = \boldsymbol{A}_j^{-1} \operatorname{vec} \left( \rho_1 \sum_{\ell=1}^{L} \left( u_{j,\ell}^{(k)} + \alpha_{j,\ell}^{(k)} \right) \boldsymbol{a}_{j,\ell}\boldsymbol{a}_{j,\ell}^* + \rho_2 \left( \boldsymbol{Z}_j^{(k)} - \boldsymbol{P}_j^{(k)} \right) - \boldsymbol{I} \right),$$

where

$$\boldsymbol{A}_j = \rho_1 \sum_{\ell=1}^{L} \operatorname{vec}\left( \boldsymbol{a}_{j,\ell}\boldsymbol{a}_{j,\ell}^* \right) \operatorname{vec}\left( \boldsymbol{a}_{j,\ell}\boldsymbol{a}_{j,\ell}^* \right)^* + \rho_2 \boldsymbol{I}.$$

Note that $\boldsymbol{A}_j^{-1}$ only needs to be calculated once throughout the entire process.

The $\boldsymbol{Z}$ update is basically the projection of a Hermitian matrix onto the PSD cone. Considering the eigen-decomposition of the Hermitian matrix $\tilde{\boldsymbol{Z}} \in \mathbb{C}^{n \times n}$:

$$\tilde{\boldsymbol{Z}} = \boldsymbol{U} \operatorname{diag}\left( \lambda_1, \cdots, \lambda_n \right) \boldsymbol{U}^*,$$

all eigenvalues are real, and the solution to

$$\underset{\boldsymbol{Z}}{\text{minimize}} \quad \frac{1}{2} \left\| \boldsymbol{Z} - \tilde{\boldsymbol{Z}} \right\|_F^2 + \mathbb{I}_+(\boldsymbol{Z})$$

is simply $\boldsymbol{U} \text{diag} \left( \max(\lambda_1, 0), \cdots, \max(\lambda_n, 0) \right) \boldsymbol{U}^*$.

Finally, the $\boldsymbol{u}$-update step in the proposed ADMM scheme requires a fast formulation of the projection onto the set $\mathcal{C}$. It is straightforward to see that program

$$\underset{\boldsymbol{u}_1, \boldsymbol{u}_2}{\text{minimize}} \quad \frac{1}{2} \sum_{j=1}^{2} \sum_{\ell=1}^{L} (u_{j,\ell} - \xi_{j,\ell})^2 + \mathbb{I}_{\mathcal{C}}(\boldsymbol{u}_1, \boldsymbol{u}_2) \tag{S.15}$$

decouples into $L$ distinct programs of the form

$$\underset{u_1, u_2}{\text{minimize}} \quad \frac{1}{2} \sum_{j=1}^{2} (u_j - \xi_j)^2 \quad \text{subject to:} \quad u_1 u_2 \ge \delta > 0, \ u_1 \ge 0. \tag{S.16}$$

Note that since the case $u_1 u_2 = 0$ leads to a trivial argument, we consider the strict inequality $\delta > 0$. In the sequel we focus on addressing (S.16), as solving (S.16) for each component $\ell$ would deliver the solution to (S.15). We proceed by forming the Lagrangian for the constrained problem (S.16)

$$l(u_1, u_2, \mu_1, \mu_2) = \frac{1}{2} \left\| \begin{pmatrix} u_1 \\ u_2 \end{pmatrix} - \begin{pmatrix} \xi_1 \\ \xi_2 \end{pmatrix} \right\|^2 + \mu_1 \left( \delta - u_1 u_2 \right) - \mu_2 u_1.$$

Along with the primal constraints, the Karush-Kuhn-Tucker optimality conditions are

$$\frac{\partial l}{\partial u_1} = u_1 - \xi_1 - \mu_1 u_2 - \mu_2 = 0, \tag{S.17}$$

$$\frac{\partial l}{\partial u_2} = u_2 - \xi_2 - \mu_1 u_1 = 0, \tag{S.18}$$

$$\mu_1 \ge 0, \quad \mu_1 \left( \delta - u_1 u_2 \right) = 0,$$

$$\mu_2 \ge 0, \quad \mu_2 u_1 = 0.$$

We now proceed with the possible cases.

**Case 1.** $\mu_1 = \mu_2 = 0$:
In this case we have $(u_1, u_2) = (\xi_1, \xi_2)$ and this result would only be acceptable when $u_1 u_2 \ge \delta$ and $u_1 \ge 0$.

**Case 2.** $\mu_1 = 0$, $u_1 = 0$:
In this case the first feasibility constraint of (S.16) requires that $\delta \le 0$, which is not a possiblity.

**Case 3.** $\delta - u_1 u_2 = 0$, $u_1 = 0$:
Similar to the previous case, this cannot happen when $\delta > 0$.

**Case 4.** $\mu_2 = 0$, $\delta - u_1 u_2 = 0$:

In this case we have $\delta = u_1 u_2$, combining which with (S.18) yields $\delta = (\xi_2 + \mu_1 u_1)u_1$, or

$$\mu_1 = \frac{\delta - \xi_2 u_1}{u_1^2}. \tag{S.19}$$

Similarly, (S.17) yields

$$u_1 = \xi_1 + \mu_1(\xi_2 + \mu_1 u_1). \tag{S.20}$$

Since the condition $\delta = u_1 u_2$ requires that $u_1 > 0$, $\mu_1$ can be eliminated between (S.19) and (S.20) to generate the following fourth order polynomial equation in terms of $u_1$:

$$u_1^4 - \xi_1 u_1^3 + \delta \xi_2 u_1 - \delta^2 = 0.$$

After solving this 4-th order polynomial equation, we pick the real root $u_1$ which obeys

$$u_1 \geq 0, \qquad \delta - \xi_2 u_1 \geq 0. \tag{S.21}$$

Note that the second inequality in (S.21) warrants nonnegative values for $\mu_1$ thanks to (S.19). After picking the right root, we can explicitly obtain $\mu_1$ using (S.20) and calculate the $u_2$ using (S.18). The resulting $(u_1, u_2)$ pair presents the solution to (S.16), and finding such pair for every $\ell$ provides the solution to (S.15).