[Reviews · NeurIPS 2018]

Reviewer 1



This paper considers blind deconvolution from phaseless measurements, i.e., recovering w and x from the measurements y = |F(w \cconv x)|, where \cconv is the circular convolution. The problem is motivated by an application in visible light communication. When w and x lie on two known random subspaces, the paper proves that w and x can be recovered (up to certain intrinsic ambiguities) via solving a convex program, provided that there are enough phaseless measurements (i.e. y is long enough) that approximately match the intrinsic degrees of freedom, i.e., length(w) + length(x). As acknowledged by the authors in Sec 1.4, the problem encompasses many difficult problems and hence the result (Theorem 1) and its implications should be interpreted carefully. For example, without the random subspace assumption on x and assuming x = 1 (all one vector or matrix), then the problem reduces to the classic Fourier phase retrieval problem, which does not warrant a provable solution yet. The randomness in the assumption seems to alleviate many aspects of difficulties associated with the original problem. As indicated in the paper, the derived convex relaxation does not seem to work well on more deterministic setting for the subspaces. As a theoretical paper that sheds light on the problem, there is a couple of interesting ideas contained in the paper. 1) The convex relaxation is novel, especially in dealing with the quadratic constraints. Exploiting positiveness of products of PSD matrices enables restriction of the hyperbola constraint into one particular branch, which in turn enables the convex relaxation. Similar strategy was used in the BranchHull paper (Aghasi et al 2017a), although there restriction was possible at the expense of additional side information on signs. 2) To show recovery performance of the convex relaxation in Eq (5), some nontrivial technicalities are dealt with. First, the constraint set is linearized near the optimal set, i.e., Lemma 1, leading to a relaxed convex program Eq (8). Second, it is shown the descent cone does not meet the subspace of nontrivial perturbation when m (# measurements) is sufficiently large, which implies that any nontrivial feasible perturbation strictly increases the objective value. Similar proof ideas also appear in the BranchHull paper. The matrix version entailed here is technically more involved. Overall, there convex relaxation and the analysis are interesting, despite the precursors that are very relevant. The practicality of the result is arguable, nevertheless. Additional Comments: * Line 44 -- 52: Another motivation the current reviewer is aware of is some ill-posed Fourier phase retrieval problem as discussed in -- Elser, Veit, Ti-Yen Lan, and Tamir Bendory. "Benchmark problems for phase retrieval." arXiv preprint arXiv:1706.00399 (2017). When the 2D signal can be modeled as a \cconv x --- say in the above paper where the data are simulated as sparse superposition of basic patterns over the image (e.g., Fig 1 there), the 2D signal is not injective if treated in the classic Fourier phase retrieval framework and hence fundamentally not recoverable. Instead, if one model the process as y = |F(a \cconv x)|, there may still be hope of recovery due to the explicit account of the structures present. * Around Line 93: It may help to formally show why the relaxed constraint set in Eq (5) is convex. * Around Line 93: The relaxation idea is very similar to the BranchHull paper. I think it is important to note this precursor here. * How easy/hard/tricky is it to obtain a result if only one of the two subspaces is random, say one of w and x is sparse with known support? This might be more relevant to practice, in view of the comments in Sec 1.4 and the other motivation I discussed above. * Page 5: What is the circled sum notation?

Reviewer 2



This paper considered the problem of blind deconvolutional phase retrieval. The main technical contribution is a novel convex relaxation that is based on a lifted matrix recovery formulation. The sample complexity for exact recovery is characterized if the two signals belong to known random subspaces (i.i.d. Gaussian entries). The characterization is derived from the Rademacher complexity estimates, which is quite interesting. The paper is well organized and well written. Despite the fact that random subspaces are a strong assumption (typical appropriate subspaces would be deterministic, including partial Discrete Cosine Transforms or partial Discrete Wavelet Transforms), I believe that this paper provides very valuable insight on this very difficult problem. I recommend the acceptance of this paper.

Reviewer 3



Summary of the paper: This paper considers a problem that combines aspects of phase retrieval and blind deconvolution. One wants to recover two vectors w and x, given phaseless measurements of the Fourier transform of the convolution of w and x. This is motivated by applications such as optical imaging using an unknown channel or filter. The authors propose a novel convex programming approach, which combines ideas from both PhaseLift (to deal with the phaseless measurements) and BranchHull (to deal with the bilinearity). They prove a recovery guarantee, with a near-optimal number of measurements, for the special case where the vectors w and x belong to known random low-dimensional subspaces. This result, although not directly relevant to practical applications (due to the assumption about random subspaces), is a step towards finding rigorous and tractable algorithms for this problem. Overall, I think this paper makes a nice conceptual contribution, by coming up with an apparently-new convex relaxation for a challenging signal recovery problem. The idea is quite simple and elegant. The proof of the recovery guarantee uses some powerful probabilistic techniques (which are not new, but are quite recent). I agree with the authors' assessment that this work is a theoretical advance, even if their method does not work that well in practice. The paper is quite clearly written. Some minor notes: In line 16, there seems to be a typo, in the formula for F[omega,t], in the exponent, where sqrt(m) should be m. In Theorem 1, can one say anything about how the constant c_t depends on t? In line 175, it may be a good idea to explain the \oplus notation.